# Load Disaggregation Based on a Bidirectional Dilated Residual Network with Multihead Attention

Yifei Shu [1], Jieying Kang [1], Mei Zhou [1], Qi Yang [1], Lai Zeng [2] and Xiaomei Yang [2,*]

1 Marketing Service Center, Ningxia Electric Power Co., Ltd., National Grid of China, Beijing 100031, China; shuyifei143313@163.com (Y.S.); kanggary1128@163.com (J.K.); nx_zm@126.com (M.Z.); y1993q0205@163.com (Q.Y.)
2 College of Electrical Engineering, Sichuan University, Chengdu 610065, China; 2021223030074@stu.scu.edu.cn
* Correspondence: yangxiaomei@scu.edu.cn

**Abstract:** Load disaggregation determines appliance-level energy consumption unintrusively from aggregated consumption measured by a single meter. Deep neural networks have been proven to have great potential in load disaggregation. In this article, a temporal convolution network, mainly consisting of residual blocks with bidirectional dilated convolution, the GeLu activation function, and multihead attention, is proposed to improve the prediction accuracy of individual appliances. Bidirectional dilated convolution is applied to enlarge the receptive field and effectively extract load features from historical and future information. Meanwhile, GeLU is introduced into the residual structure to overcome the "dead state" issue of traditional ReLU. Furthermore, multihead attention aims to improve the prediction accuracy by giving different weights according to the importance of different-level load features. The proposed model is validated using the REDD and UK-DALE datasets. Among six existing neural networks, the experimental results demonstrate that the proposed algorithm achieves the least average errors when disaggregating four appliances in terms of mean absolute error (MAE) and signal aggregate error (SAE), respectively, reduced by 22.33% and 60.58% compared with the model with the second-best performance on the REDD dataset. Additionally, the proposed algorithm shows superior results in identifying the on/off state in four appliances from the UK-DALE dataset.

**Keywords:** load disaggregation; residual structure; bidirectional dilated convolution; multihead attention; GeLU

## 1. Introduction

Load disaggregation, treated as a regression task in nonintrusive load monitoring (NILM) [1], is used to estimate appliance-level energy consumption from the aggregated consumption of a household measured by a single meter. Although employing a submeter for each appliance, or intrusive load monitoring, could precisely provide its energy consumption information, it would bring high installation costs and installation complexity. In contrast, NILM is a more economical and effective method that requires only one measurement point, and the appliance-level energy consumption can be obtained by using disaggregation algorithms from the total household energy. With the help of nonintrusive load disaggregation, consumers can check and adjust the energy usage of their home appliances, and electric power companies can effectively plan power supply strategies and incentive policies to reduce energy wastage [2].

Over the past few years, many studies have examined load disaggregation algorithms using various technologies, such as mathematical optimization, machine learning, and deep learning methods. Some optimization methods, such as particle swarm optimization [3], sparse optimization [4], quadratic programming [5], and the combination of optimization and factor hidden Markov model methods [6,7], attempt to find the best combination of appliances by minimizing the difference between aggregated consumption and the sum

of appliance-level consumption. Commonly, these methods have high computational complexity, making them unsuitable for real-time applications.

With the advent of big data, machine learning methods apply stationary or nonstationary signal processing algorithms to extract time-domain or transformation-domain features from a large quantity of data [8–10], and then pattern recognition, both supervised and unsupervised, is used to implement load disaggregation, such as clustering [11] and the AdaBoost algorithm [12]. The experiments mentioned above have verified that the disaggregation accuracy mainly depends on the manually extracted features, which are difficult to identify due to the complexity of appliance operation.

Benefiting from the good performance of deep learning in computer vision, speech recognition, and natural language processing, deep neural networks (DNNs) have been applied to improve the accuracy and practicality of load disaggregation. Compared to classical machine learning methods, DNN-based load disaggregation approaches can automatically extract latent features, rather than carrying out manual work from measured electrical signals, and they can effectively construct the nonlinear relationship between the inputs and outputs of the related problem. To extract the time-series data features, a long short-term memory (LSTM) network [13] is proposed to perform nonlinear regression tasks using sequence-to-sequence (S2S) learning, where a network outputs a prediction sequence with the same length as an input sequence. After this, other improved recurrent neural networks (RNNs), such as bidirectional LSTM (Bi-LSTM) [14] and gated recurrent units (GRU) [15], were proposed to improve the performance of load disaggregation. Considering that RNN architectures cannot perform parallel computation, a variety of convolutional neural networks (CNNs) are used in load disaggregation via S2S learning [16] or sequence-to-point (S2P) learning [17]. In S2P learning, the network predicts the midpoint of the output signal rather than the whole sequence. To obtain the high-level features of appliance-level data, deep denoising autoencoders [18] and fully convolutional networks (FCNs) [19] have been proposed. Commonly, CNNs are time-independent, while appliance-level power consumption has the characteristics of time dependency, especially for multistate appliances. Thus, based on RNN and CNN, various improved network architectures have been proposed in this research field. For example, a lightweight algorithm combining a deep CNN and a KNN classifier is used to identify several appliances with computational efficiency in [20], and a residual network18 with squeeze and excitation is used to avoid the training difficulty problem of classical CNN in [21]. In addition, to address the data-hungry issue of DNN-based algorithms, transfer-learning-based disaggregation approaches have been utilized to transform the features learned by DNNs using one appliance into those of another appliance [22,23]. More related reviews can be found in [24–26].

In this paper, we focus on predicting appliance-level power consumption from aggregated consumption by combining the usage of multihead self-attention with a bidirectional temporal convolution network (TCN), denoted as the Attention-bitcn model, and then further identify the on/off state of the appliance by comparison with an appliance threshold. The proposed model employs a residual structure to alleviate gradient disappearance and explosion. By employing bidirectional dilated convolution, the proposed model not only enlarges the receptive field but also makes predictions more accurately using the previous and future information, since appliance operation is often noncausal. For example, a washing machine has predefined operational cycles, e.g., the washing run always precedes the dryer cycle. Different from the bidirectional dilated residual network in BitcnNILM [27], which utilizes ReLU as an active function followed by dropout in the residual block, we use the Gaussian error linear unit (GeLU) instead of ReLU, where GeLU overcomes the "dead state" issue of ReLU and integrates the dropout and ReLU properties, simplifying the residual block and making it compact. Considering that there are differences in the importance of the multilevel load features extracted from the multiple residual blocks with different dilated factors, multihead self-attention is employed after the dilated residual network to enable the model to focus on the most important load features from the different-level feature maps and provide different weights according to the load feature importance. After

load disaggregation is implemented using the proposed model, the on/off state of the appliance is identified by comparing the decomposed result with a preset power threshold provided in [27].

To summarize, the main contributions of this work are as follows.

1.  An architecture combining a bidirectional TCN with multihead self-attention is constructed and trained to implement nonintrusive load disaggregation.
2.  Bidirectional dilated convolution within bidirectional TCN is employed to maximize the receptive field and improve the prediction from previous and future information; meanwhile, GeLU, integrating the properties of dropout and ReLU, is used as an active function to make the residual block compact.
3.  Multihead self-attention within the proposed algorithm is utilized to capture the correlations of different-level load features.
4.  The REDD and UK-DALE datasets are used to validate the proposed algorithm, which achieves the least average errors for the disaggregation of four appliances in the REDD dataset and shows superior results in identifying the on/off states of four appliances in the UK-DALE dataset.

The structure of the present paper is as follows. Section 2 briefly introduces the problem formulation of load disaggregation. Section 3 describes the proposed algorithm, including the residual structure of bidirectional dilated convolution, multihead self-attention, and training. Section 4 introduces the experimental results, including the datasets, evaluation indices, comparative analysis of the proposed algorithm, and ablation experiments. Conclusions are drawn in Section 5.

## 2. Problem Formulation

The NILM problem is to monitor the operating state of an individual appliance by decomposing the total energy consumption into individual consumption, where the total or individual consumption can be represented using any electrical variables, such as current, voltage, active power, reactive power, or a vector of all of them. In this study, the active power is considered to be the electrical variable. At time $t$, the aggregate active power $y(t)$ is given by the sum of the individual power and can be expressed as follows:

$$y(t) = \sum_{n=1}^{N} x_n(t) + \varepsilon(t), \tag{1}$$

where $x_n(t)$ denotes the active power of appliance $n$, $N$ denotes the total number of appliances, and $\varepsilon(t)$ denotes the Gaussian noise measurement.

Accordingly, the energy disaggregation can be described to establish the nonlinear function $f_n(\cdot)$, given by

$$x_n(t) = f_n(y(t)), \tag{2}$$

which maps a sequence of total consumption $y$ into a sequence of individual consumption $x_n$ with the same length.

## 3. The Proposed Algorithm

To obtain the nonlinear mapping function $f_n(\cdot)$ in (2), we construct a deep learning network, as shown in Figure 1, which mainly consists of one standard convolution layer, eight residual blocks with bidirectional dilated convolution, multihead attention, and a dense layer. First, the sliding-window data of the aggregate power are fed into the standard convolution layer to extract low-level load features. Furthermore, eight residual blocks are used to extract higher-level load features. The output of each residual block is connected with the next stack, and its output is concatenated to construct the representation of different-level load features. Multihead self-attention is further used to capture the correlations of different-level load features. To reduce the overfitting issue during training, the dropout layer is implemented after the attention weights are obtained in multihead self-attention. Finally, the dense layer transforms the feature vector into a target point with S2P learning. After the proposed network is trained to implement load disaggregation, the

on/off state of the appliance is identified by comparing it with a preset appliance threshold. More details on the core components, i.e., the residual block with bidirectional dilated convolution, multihead attention, and network training, are presented in the following.

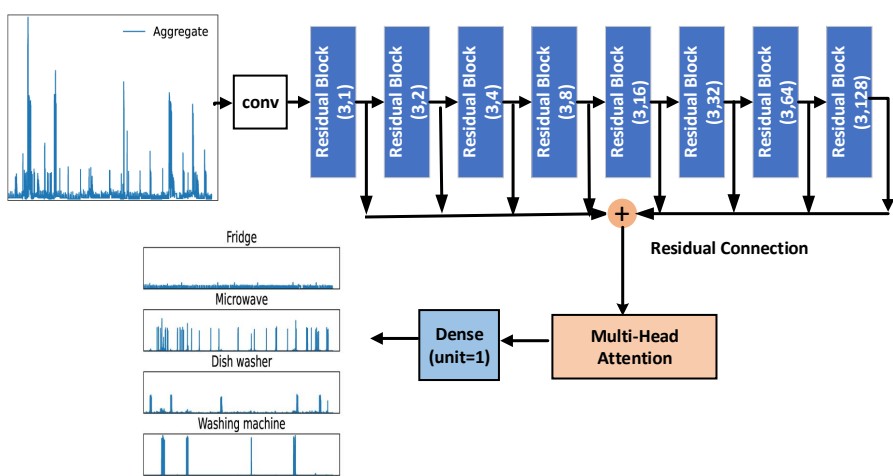

**Figure 1.** The proposed architecture for load disaggregation, where (*a*, *b*) denote the filter length and dilated factor in the residual block, respectively.

### 3.1. Bidirectional Dilated Convolution

To effectively extract load features with long-term dependence and improve the load disaggregation accuracy, bidirectional dilated convolution with information from previous and future directions is utilized to overcome the causal dilated convolution limitation with information from only a single previous direction.

Commonly, causal dilated convolution is formed by combining causal convolution and dilated convolution, where the causal convolution is used to directly predict the value of the next layer at time *t* from the value of the previous layer at time *t* and before time *t*, not relying on the information from the further time steps, as shown in Figure 2a, while the dilated convolution is used to expand the receiving field, not relying on the increase in the convolution kernel or the convolution depth. A stack of causal dilated convolutions with 4 layers and a filter length of 2 is shown in Figure 2b, where the convolution filter is applied over an area larger than its length using interval sampling with different dilated factors (denoted as *d*). When $d = 1$, each time point needs to be sampled to calculate the inner product with the convolution kernel; when $d = 2$, there will be one time point skipped between every two points to execute the convolution operation, and so on. Generally, *d* is increased to $2^{j-1}$, where *j* denotes the number of convolutional layers. As the number of layers increases, the greater the dilation factor *d*, the larger the receiving field. For a 1-D time-series input $f \in \Re^n$ and convolution kernel function *g*, the causal dilated convolution operation with *d* on *x* is defined as

$$F(t) = (f * dg)(t) = \sum_{i=0}^{k-1} g(i) f_{t-d \cdot i} \tag{3}$$

where $*$ denotes the dilated convolution operator, *k* denotes the convolutional kernel size, and $t - d \cdot i$ denotes the past direction information.

However, the causal dilated convolution achieves only a single directional prediction, and it is not comprehensive enough to extract the load features. For load disaggregation in the NILM problem, future samples are generally useful in improving predictions, which can be seen from the success of bidirectional RNNs. Thus, to improve the accuracy, bidirectional dilate convolution is proposed to predict $F(t)$ by utilizing both historical information ($[f_{t-di}, ..., f_t]$) and future information ($f_{t+1}, f_{t+2}, f_{t+di}$), defined as

$$F(t) = (f * dg)(t) = \sum_{i=-(k-1)}^{k-1} g(i) f_{t+d \cdot i}, \tag{4}$$

where $t + d \cdot i$ represents the past-directional information when $i < 0$ and the future-directional information when $i > 0$. The architecture of bidirectional dilated convolutions with 4 layers and a filter length of 3 is shown in Figure 2c.

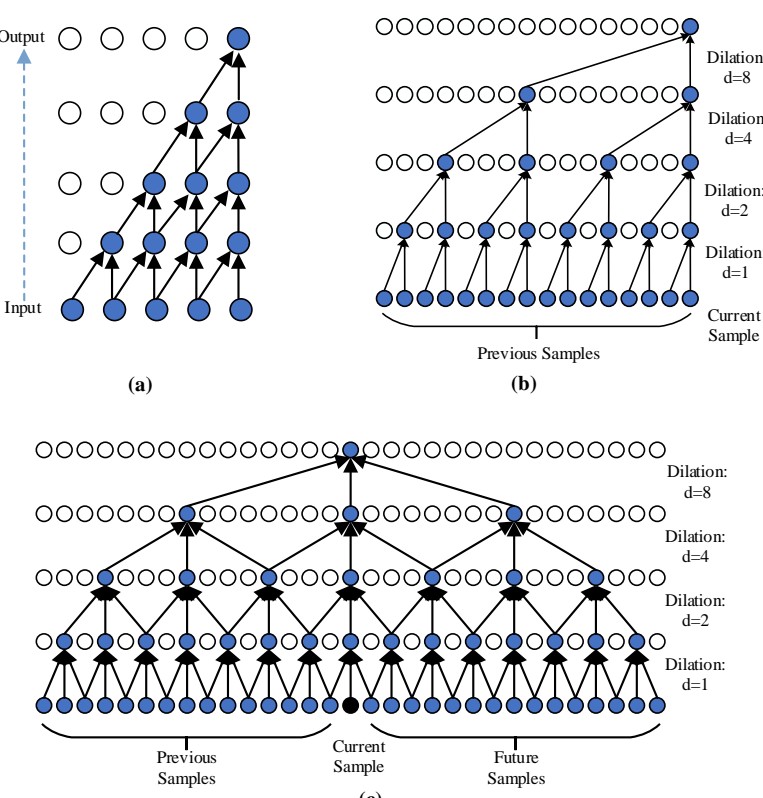

**Figure 2.** Three types of stacked convolutions [27]. (**a**) Causal convolution; (**b**) dilated causal convolution; (**c**) bidirectional dilated convolution.

*3.2. The Residual Block*

To effectively improve the training efficiency and largely handle the gradient disappearance and gradient explosion problems, the residual block is applied in the proposed model and constructed by residual connection and two residual units, as shown in Figure 3. Each of the convolution units contains bidirectional dilated convolution, batch normalization (BN), and GeLU activation [28,29]. As conventional deep networks do, BN follows bidirectional dilated convolution to speed up and stabilize the deep neural network training.

Different from ReLU, which is commonly used as an active function in TCNs or CNNs, we utilize GeLU as the active function in the residual block. The reason can be explained by two aspects. (1) ReLU would cause the issue of "dead neurons". When the value of the feature input is smaller than zero, the ReLU output is always zero and thus its first derivative is zero, which leads to some neurons being dead. Because dead neurons give zero activation, the weight parameters cannot be updated in future feature data points. This issue of ReLU hinders learning and makes the network performance poor. (2) Dropout is commonly followed by ReLU to avoid overfitting in TCNs or CNNs, while GeLU merges the properties of dropout and ReLU [28], where the input is multiplied by zero or one. The zero–one mask is stochastically determined and also dependent upon the input $z$ and simulated by Bernoulli distribution $\phi(z)$, where $\phi(z) = P(Z \leq z)$, $Z \sim N(0,1)$. The formula for GeLU is

$$GeLU(z) = xP(Z \leq z) = z\phi(z), \tag{5}$$

In application, GeLU is commonly approximated by

$$GeLU(z) = 0.5z\left(1 + tanh[\sqrt{2/\pi}(z + 0.044715z^3)]\right) \tag{6}$$

More details of GeLU are provided in [28].

The residual connection is chosen for each residual block to alleviate the difficulty of training deeper networks. The residual connection is achieved by the jump joining and addition operation, as shown in Figure 3. Let $I_n$ be the input features of the $n$th residual block, and its output $I_{n+1}$ can be expressed as

$$I_{n+1} = F(I_n) + I_n, \tag{7}$$

where $I_{n+1}$ is the output of the current residual block, used as the input feature for the next residual block, and $F(\cdot)$ is the output of the second residual unit within the current residual block.

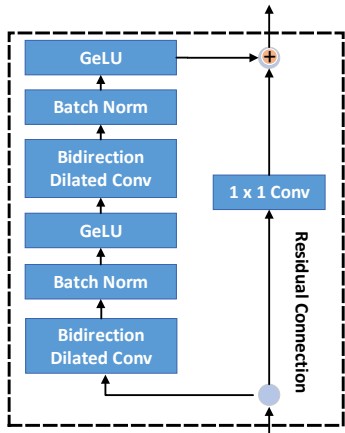

**Figure 3.** Residual block.

### 3.3. Multihead Self-Attention

Multihead self-attention allows the model to capture correlations and the weighted combination between different-level load features extracted from eight residual blocks with different dilated factors [30,31]. After the outputs of the eight residual blocks are concatenated, multihead self-attention is employed to assign different weights according to the importance of the input features.

The attention mechanism adopts the "scaled dot-product attention" by operating on three vectors, i.e., query ($Q$), key ($K$), and value ($V$). In scaled dot-product attention, as shown in Figure 4a, the dot products of the queries with all keys are computed to estimate the significance of each key. Then, the results of the dot product are divided by $\sqrt{d}$, where $d$ denotes the dimensions of $K$ and $Q$. Next, a softmax function is applied to obtain the weights, which represent the relative importance of each pair of key values to the particular query. Finally, each attention weight is multiplied by the corresponding value to obtain the output, and the corresponding attention function is given by

$$\text{Attention}(Q,\ K,\ V) = \text{softmax}\left(\frac{QK^{\text{T}}}{\sqrt{d}}\right)V. \tag{8}$$

The multihead attention can analyze the input features from various aspects by performing a single attention function $h$ times, which takes different linear projections of queries, keys, and values as inputs, as shown in Figure 4b. Here, $h$ denotes the number of heads in multihead self-attention. Then, the outputs of the attention mechanisms are

concatenated to further perform the linear projection. Correspondingly, multihead attention is formulated as

$$\text{MultiHead}(Q, K, V) = \text{concat}(H_1, H_2, ..., H_h)W^O$$
$$H_i = \text{Attention}(QW_i^Q, KW_i^K, VW_i^V) \tag{9}$$

where $(W^O, W_i^Q, W_i^K, W_i^V)$ are the linear projection matrices, $H_i$ denotes the output of a single attention function, and $h = 6$ is set in our experiments.

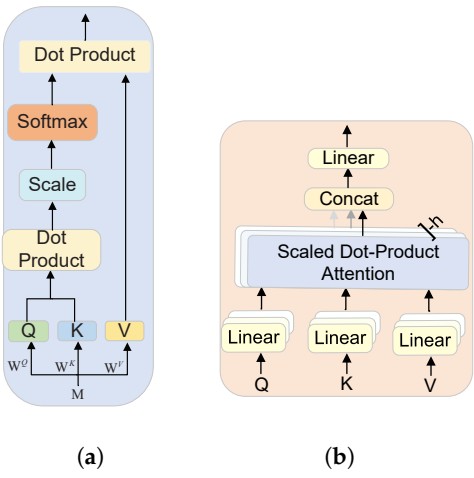

**(a)**　　　　　**(b)**

**Figure 4.** The scaled dot-product attention block is shown on the left (**a**), and the multihead attention block is shown on the right (**b**).

### 3.4. Training of the Proposed Network

To eliminate the influence of different units or scales during training and testing, all data $D(t)(\{x_n(t), y(t)\} \subset D(t))$ at time $t$ from each appliance or mains are normalized as

$$D(t) = \frac{D(t) - \overline{D}}{\sigma}, \tag{10}$$

where $\overline{D}$ and $\sigma$ denote the mean and standard deviation of the power, respectively.

To train the proposed network, the sliding-window mechanism is applied to divide the whole sequence of aggregate power ($y(t)$) and the sequence of individual appliances ($x_n(t)$) into windows of fixed length $L$ and overlapped by $P < L$ samples. We define a bag $j$ as the $j$-th window of $y(t)$ and $x_n(t)$ as follows:

$$y_j = [y(j(L-P)), ..., y(j(L-P) + L - 1)],$$
$$x_{n,j} = [x_n(j(L-P)), ..., x_n(j(L-P) + L - 1)], \tag{11}$$

where $x_{n,j}$ denotes the ground-truth data.

Given a set of annotated bags $T$, denoted as

$$T = \{(y_1, x_{1,1}, .., x_{N,1}), ...., (y_j, x_{1,j}, .., x_{N,j})\}, \tag{12}$$

the network is trained using a loss function, calculated as the mean squared error between the related prediction and the target. Considering a minibatch of $J$ bags and a generic appliance, the loss function is given by

$$\mathcal{L}_s = \frac{1}{NJL} \sum_{j=0}^{J-1} \sum_{l=0}^{L-1} \sum_{n=0}^{N} [x_n(j(L-P) + l) - \widehat{x}_n(j(L-P) + l)]^2 \tag{13}$$

where $\widehat{x}_n$ is the predicted power of $x_n$ for the individual appliance. Once $\widehat{x}_n$ is obtained, the on/off state of the individual appliance is predicted by comparison with the corresponding

threshold, where the threshold and *L* are set the same as in [27] to fairly evaluate the performance of the proposed model.

## 4. Experimental Results

The performance of the proposed Attention-bitcn model on load disaggregation and on/off state identification is evaluated using a public REDD [32] dataset in this section. The proposed model is implemented on the TensorFlow and Keras frameworks, and the training is conducted by the Adam optimizer, where two hyperparameters are set as $\beta_1 = 0.9$ and $\beta_2 = 0.999$. Additionally, the batch size, learning rate, dropout rate, and epochs are set to 128, $10^{-3}$, 0.3, and 20, respectively. Considering that the proposed algorithm belongs to the type of S2P learning, the performance of the proposed algorithm is verified by comparing it with the related S2P models, i.e., CNN (S2P) [17], FCN [19], BitcnNILM [27]. Meanwhile, the proposed approach is compared with two S2S learning models, i.e., LSTM and Bi-LSTM are both constructed using the corresponding modules provided by Keras directly, where the number of hidden units is set to 128 in the two S2S models. For a fair comparison, the epochs in FCN, BitcnNILM, LSTM, and Bi-LSTM are set as 20, as in the proposed model. The epochs are set to 2 in the CNN (S2P) provided. Furthermore, an ablation experiment is performed to evaluate the effectiveness of the multihead attention and GeLU modules. All the experiments described in this article were implemented using an NVIDIA RTX3060 GPU with 12 GB of RAM.

### 4.1. Dataset

Two public datasets, REDD [32] and UK-DALE [33], were used in the following experiments. The REDD dataset [32] contains high-frequency (15 Hz) and low-frequency data of power consumption recorded in 3 s for six households. Among these, the low-frequency data from Homes No. 2 to No. 6 are used for training, and the data from Home No. 1 are used for testing. The UK-DALE dataset [33] contains power consumption data from five houses, where the aggregated and individual appliance power consumption were recorded every 6 s. Similar to the REDD dataset, four appliances, including a microwave, fridge, dishwasher, and washing machine, are also considered in the UK-DALE dataset. The data from Home No. 2 are divided by 6:2:2 to use as the training set, validation set, and testing set, respectively. To evaluate the proposed model and perform the comparison with other NILM models, four appliances, i.e., a microwave, fridge, dishwasher, and washing machine, are utilized in our works, since the selected appliances consume a large proportion of power energy in a household and have generally representative but different power patterns; for example, the fridge is a two-state appliance, while the dishwasher and washing machine are multi-state appliances.

### 4.2. Evaluation Metrics

To evaluate the accuracy of the proposed model in load disaggregation, the mean absolute error (MAE) and signal aggregate error (SAE) are selected as the evaluation indices. MAE measures the average error in the predicted power consumption of a single appliance decomposed at each moment *t* from the ground truth, defined as

$$\text{MAE} = \frac{1}{T} \sum_{t=1}^{T} |\widehat{x}_t - x_t|, \tag{14}$$

where $\widehat{x}_t$ denotes the predicted value of the individual appliance, decomposed from the total consumption; $x_t$ denotes the ground-truth value; and *T* denotes the number of time points. Correspondingly, the all-over MAE is calculated as the average value of the MAE for all four appliances.

SAE measures the total error of power consumption within a period, defined as

$$\text{SAE} = \frac{\widehat{r} - r}{r}, \tag{15}$$

where $\hat{r}$ and $r$ denote the predicted value and the ground-truth value of daily power consumption, respectively. Similarly, the all-over SAE is calculated as the average value of the SAE for all four appliances.

To evaluate the performance of the proposed model's on/off state identification of individual appliances, since state identification can be considered a classification task, four metrics, i.e., recall (R), precision (P), $F_1$ score, and accuracy (A), are used. They are calculated as

$$
\begin{aligned}
R &= \frac{TP}{TP + TN}, & P &= \frac{TP}{TP + FP}, \\
A &= \frac{TP + TN}{TP + TN + FP + FN}, & F_1 &= 2 \times \left( \frac{P \times R}{P + R} \right),
\end{aligned}
\tag{16}
$$

where TP (true positive) denotes the number of instances with an "on" state that are correctly predicted as "on"; TN (true negative) is the number of instances whose predicted and ground-truth states both are "off"; FP (false-positive) is the number of instances with an "off" state that are incorrectly predicted as "on"; FN (false-negative) is the number of instances with an "on" state that are incorrectly predicted as "off".

### 4.3. Experimental Results

4.3.1. Experiments on REDD Dataset

The proposed model implements load disaggregation for four appliances from Home No. 1 in the REDD dataset after the parameters of the Attention-bitcn model are trained. The corresponding results are shown in Figure 5. From Figure 5, it can be seen that the load curves predicted by the proposed Attention-bitcn model for microwaves, dishwashers, and washing machines are consistent with the ground-truth load curves. However, for the fridge, the proposed model appears unable to predict the peak of the load curve well. There are visual differences between the predicted curve and the ground-truth curve because the sample points are too dense. After enlarging the partially decomposed curve of the fridge within [160,000, 170,000] time samples, as shown in Figure 6, we observe that the predicted power consumption is also consistent with the actual load, except for the sudden changes in load corresponding to the lines in orange, which can be ignored because of the small number of sudden changes, as noted in [27].

Next, the disaggregation performance of the proposed model is compared with those of the other five models in terms of the MAE and SAE indices. As shown in Figure 7, the proposed Attention-bitcn model achieves the best disaggregation performance for three appliances, i.e., microwaves, dishwashers, and washing machines. Compared with Bitcn-NILM, with the second-best performance, the proposed Attention-bitcn model improves more significantly for the microwave and dishwasher, with the MAE decreasing by 4.2572 watts and 4.7295 watts, respectively. In the decomposition model of washing machines, the decomposition accuracy of the proposed network algorithm is lower than that of Bitcn-NILM. In comparison with the two S2S models (i.e., LSTM and Bi-LSTM), the proposed model has a lower MAE or higher decomposition accuracy on four appliances. Meanwhile, the overall MAE of the proposed algorithm is reduced by 0.5301 watts, 9.4675 watts, 10.0107 watts, 16.7448 watts, and 12.7838 watts, compared with BitcnNILM, CNN (S2P), FCN, LSTM, and Bi-LSTM, respectively. Additionally, from the perspective of the SAE index, the proposed Attention-bitcn model also achieves comparable and even superior performance with a reduced SAE for specific appliances and overall. As a whole, the proposed algorithm reduces the overall MAE and SAE by 22.33% and 60.58% compared with the BitcnNILM model with the second-best performance, respectively.

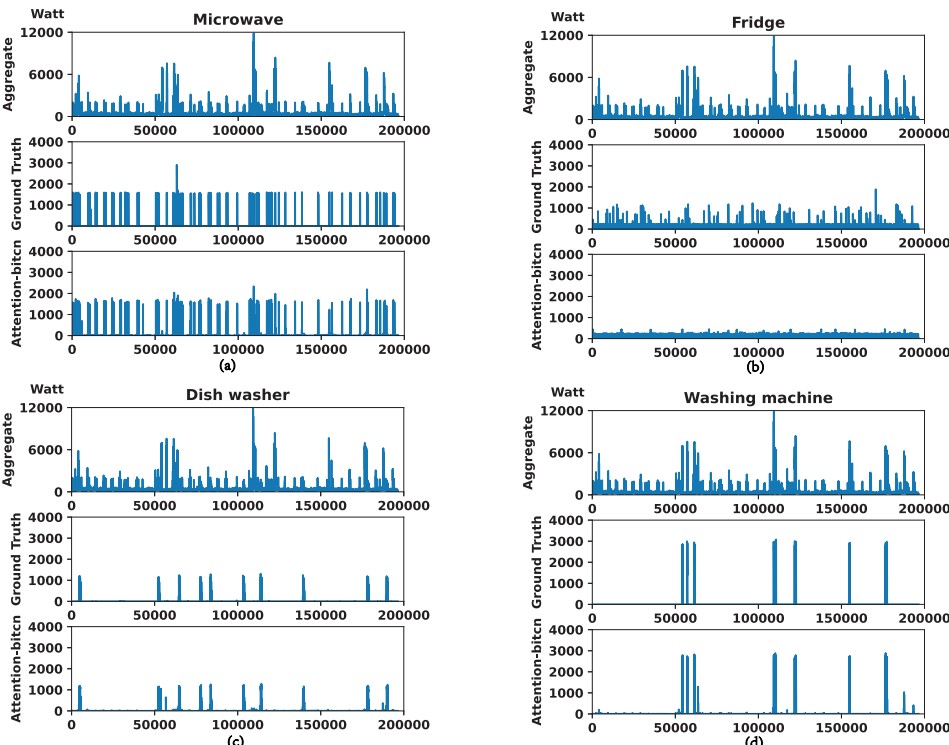

**Figure 5.** Disaggregated results for four appliances in proposed model on REDD dateset. (**a**) Microwave; (**b**) fridge; (**c**) dishwasher; (**d**) washing machine.

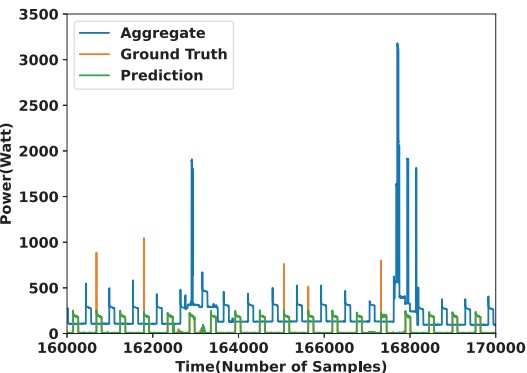

**Figure 6.** Enlarged partial curve of load disaggregation for fridge.

Furthermore, the performance indices of the on/off state identification of individual appliances, obtained from the six algorithms, are shown in Table 1, where the best metric values are shown in bold. It can be observed in Table 1 that, among all the compared models, the proposed algorithm yields the best metric values for dishwashers. Even if the metric values of the proposed algorithm are slightly inferior to the best indicator values for the other three appliances, there is little difference from the best metric values. This implies that the proposed algorithm provides overall good performance for the on/off state identification of appliances.

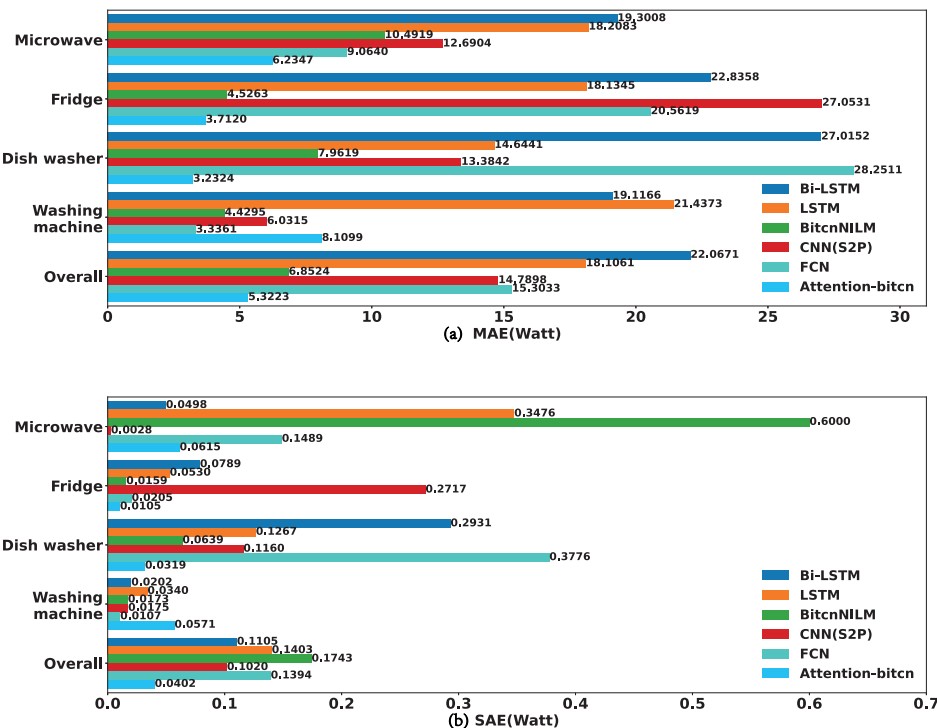

**Figure 7.** In the REDD dataset, the performance of load disaggregation for six comparison models. (**a**) MAE for each appliance and overall MAE; (**b**) SAE for each appliance and overall SAE.

**Table 1.** In the REDD dataset, comparisons of on/off state identification of the individual appliances.

| | Microwave | | | | Fridge | | | |
|---|---|---|---|---|---|---|---|---|
| | **P** | **R** | **A** | **F1** | **P** | **R** | **A** | **F1** |
| Attention-bitcn | 0.9110 | 0.9546 | 0.9982 | **0.9323** | 0.9917 | **0.9972** | 0.9972 | 0.9944 |
| CNN(S2P) [17] | 0.7730 | **0.9933** | 0.9988 | 0.8694 | 0.9551 | 0.9240 | 0.9467 | 0.9392 |
| FCN [19] | 0.7857 | 0.9742 | 0.9963 | 0.8699 | 0.8126 | 0.9818 | 0.9392 | 0.8892 |
| BitcnNILM [27] | **0.9354** | 0.9275 | **0.9995** | 0.9314 | **0.9974** | 0.9965 | **0.9973** | **0.9969** |
| LSTM | 0.7439 | 0.9680 | 0.9873 | 0.8547 | 0.7733 | 0.9727 | 0.9219 | 0.8616 |
| Bi-LSTM | 0.4084 | 0.9873 | 0.9423 | 0.5778 | 0.7531 | 0.9703 | 0.9131 | 0.8480 |
| | **Dishwasher** | | | | **Washing Machine** | | | |
| | **P** | **R** | **A** | **F1** | **P** | **R** | **A** | **F1** |
| Attention-bitcn | **0.8262** | **0.9921** | **0.9913** | **0.9016** | 0.5525 | 0.9950 | 0.9901 | 0.7105 |
| CNN(S2P) [17] | 0.4169 | 0.9817 | 0.9443 | 0.5853 | 0.5589 | **1.0000** | 0.9903 | 0.7170 |
| FCN [19] | 0.1613 | 0.9967 | 0.7910 | 0.2776 | **0.6417** | 0.9917 | **0.9931** | **0.7818** |
| BitcnNILM [27] | 0.3897 | 0.9859 | 0.9377 | 0.5586 | 0.4450 | 0.9963 | 0.9848 | 0.6152 |
| LSTM | 0.4061 | 0.9716 | 0.9420 | 0.5728 | 0.4178 | 1.0000 | 0.9828 | 0.5864 |
| Bi-LSTM | 0.3689 | 0.3274 | 0.9844 | 0.3470 | 0.4007 | 1.0000 | 0.9817 | 0.5721 |

### 4.3.2. Experiments on UK-DALE Dataset

Similar to the experiments on the REDD dataset, the proposed algorithm is trained and tested on the UK-DALE dataset, and the disaggregation results for four appliances are shown in Figure 8. For microwaves, dishwashers, and washing machines, the disaggregated load curves in Figure 8 show that the proposed algorithm achieves satisfactory results, whereas it is unsatisfactory for the fridge, similar to that of the REDD dataset. After some investigation, we also find that the error is mainly caused by the abrupt power changes of the two-state fridge, shown in the vertical lines in Figure 8b.

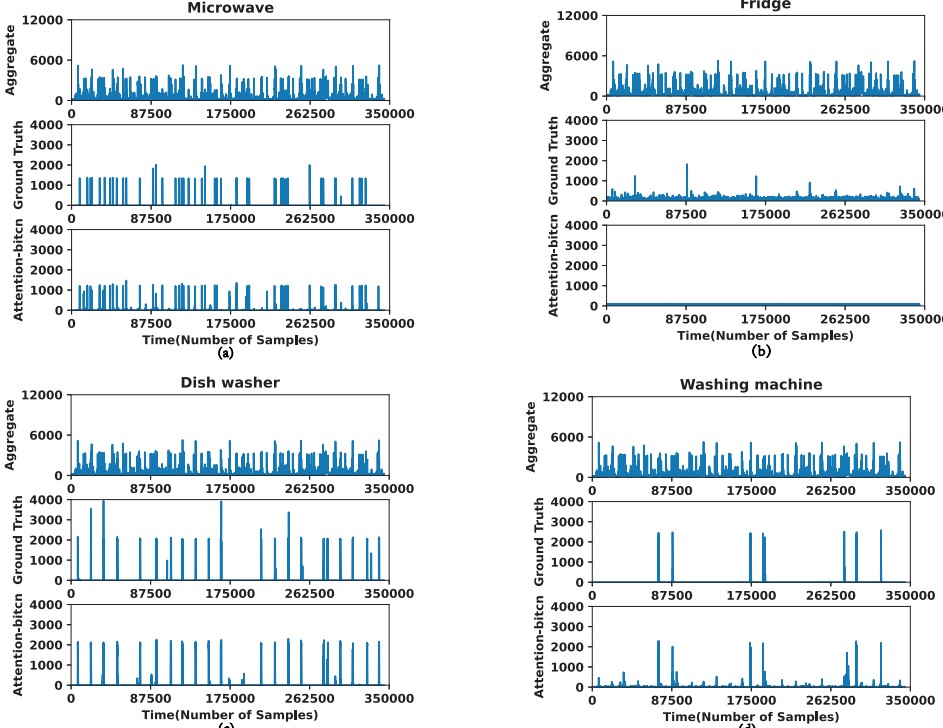

**Figure 8.** Disaggregated results for four appliances in proposed model on UK-DALE dataset. (**a**) Microwave; (**b**) fridge; (**c**) dishwasher; (**d**) washing machine.

Furthermore, the MAE and SAE indices are calculated statistically for the proposed algorithm to evaluate the performance of disaggregation. After the other five algorithms are separately trained and tested on the UK-DALE dataset, we plot the MAE and SAE indices of all the algorithms in Figure 9 and report the on/off state identification of four appliances in Table 2. On the UK-DALE dataset, the proposed algorithm achieves the second-best result in the MAE and SAE indices, whereas there is a small gap between the proposed algorithm and BitcnNILM, which has the lowest MAE and SAE. Moreover, the proposed algorithm shows better on/off identification indices than BitcnNILM on the UK-DALE dataset.

Combining the above results, we find that, for the fridge, with a relatively short operation cycle, the proposed algorithm achieves better decomposition performance on the REDD dataset than the UK-DALE dataset, where the sampling frequency of REDD is higher than that of UK-DALE. The power sequences with a high time resolution are crucial for the proposed algorithm to capture the feature information. Additionally, for the dishwasher, with complex working states and longer operation cycles, the proposed algorithm achieves superior results in on/off state identification on the two datasets.

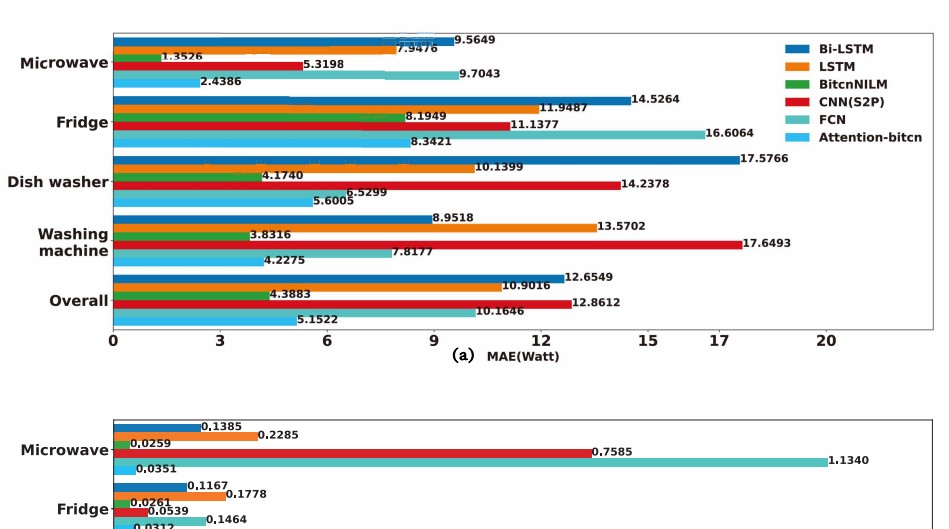

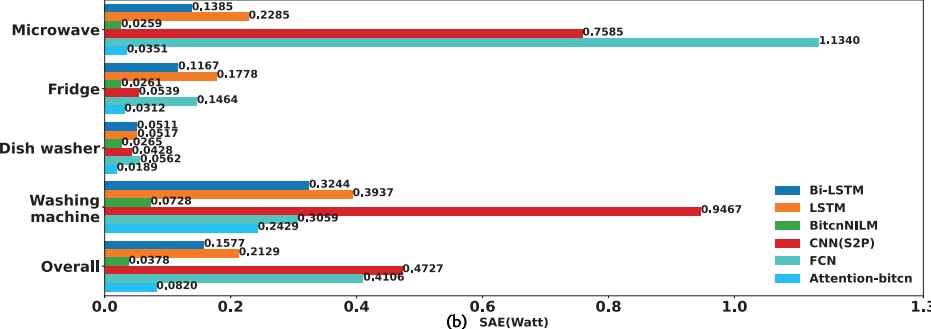

**Figure 9.** In UK-DALE dataset, the performance of load disaggregation for six comparison models. (**a**) MAE for each appliance and overall MAE; (**b**) SAE for each appliance and overall SAE.

**Table 2.** In UK-DALE dataset, comparisons of on/off state identification of the individual appliances.

| | Microwave | | | | Fridge | | | |
|---|---|---|---|---|---|---|---|---|
| | **P** | **R** | **A** | **F1** | **P** | **R** | **A** | **F1** |
| Attention-bitcn | **0.9003** | 0.8857 | **0.9993** | **0.8929** | 0.8669 | 0.9451 | **0.9306** | 0.9043 |
| CNN(S2P) [17] | 0.6864 | **0.9693** | 0.9984 | 0.8037 | 0.8009 | **0.9530** | 0.9014 | 0.8703 |
| FCN [19] | 0.3225 | 0.9113 | 0.9932 | 0.4764 | 0.7891 | 0.9422 | 0.8925 | 0.8589 |
| BitcnNILM [27] | 0.8662 | 0.9002 | 0.9992 | 0.8828 | **0.8813** | 0.9512 | 0.9386 | **0.9150** |
| LSTM | 0.5843 | 0.8694 | 0.9966 | 0.7472 | 0.7031 | 0.7936 | 0.7318 | 0.8042 |
| Bi-LSTM | 0.5537 | 0.8521 | 0.9942 | 0.7017 | 0.6799 | 0.8081 | 0.8014 | 0.7385 |

| | Dishwasher | | | | Washing Machine | | | |
|---|---|---|---|---|---|---|---|---|
| | **P** | **R** | **A** | **F1** | **P** | **R** | **A** | **F1** |
| Attention-bitcn | **0.8130** | **0.9627** | 0.9843 | 0.7771 | **0.8442** | 0.8860 | **0.9970** | **0.8646** |
| CNN(S2P) [17] | 0.8009 | 0.9530 | 0.9014 | **0.8703** | 0.2955 | 0.9492 | 0.9007 | 0.1736 |
| FCN [19] | 0.3383 | 0.9283 | 0.9404 | 0.4959 | 0.1167 | **0.9831** | 0.9178 | 0.2086 |
| BitcnNILM [27] | 0.7584 | 0.9139 | **0.9881** | 0.8289 | 0.7693 | 0.9341 | 0.9962 | 0.8437 |
| LSTM | 0.4470 | 0.8751 | 0.8906 | 0.5349 | 0.5880 | 0.8481 | 0.9734 | 0.7120 |
| Bi-LSTM | 0.6083 | 0.9393 | 0.9418 | 0.6942 | 0.5695 | 0.8905 | 0.8881 | 0.7039 |

*4.4. Ablation Study*

To evaluate the effectiveness of GeLU and the attention module in the proposed network, we perform an ablation experiment by removing the corresponding module separately from the proposed network model.

The model with the removal of GeLU, denoted as "w/o GeLU", refers to the replacement of GeLU with ReLU and BN in the residual block while retaining multihead attention, while the model with the removal of attention, denoted as "w/o Attention", refers to the removal of multiattention while keeping GeLU. As the above experiments are performed, these models are trained and tested on the REDD dataset, and the corresponding MAE and

SAE indices are shown in Figure 10. The proposed Attention-bitcn model yields the lowest MAE and SAE for microwaves and dishwashers compared with "w/o GeLU" and "w/o Attention". Overall, the absence of GeLU or the attention module increases the overall MAE and SAE, which implies that GeLU and the attention module play a role in improving the performance of the proposed algorithm in load disaggregation. Furthermore, the effectiveness of GeLU and the attention module can be seen from the indicator values of on/off state identification, as shown in Table 3.

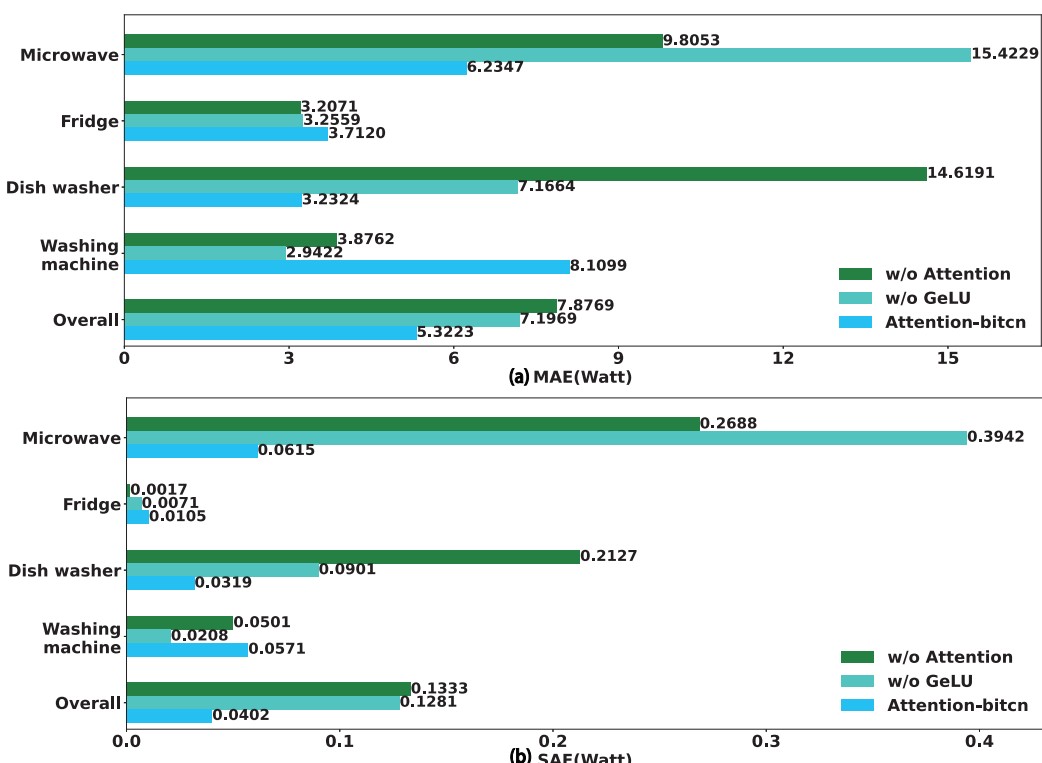

**Figure 10.** Performance of load disaggregation with/without GeLU or attention module. (**a**) MAE for each appliance and overall MAE; (**b**) SAE for each appliance and overall MAE.

**Table 3.** Comparisons of on/off state identification of the individual appliances in ablation experiment.

| | Microwave | | | | Fridge | | | |
|---|---|---|---|---|---|---|---|---|
| | **P** | **R** | **A** | **F1** | **P** | **R** | **A** | **F1** |
| Attention-bitcn | **0.9110** | 0.9546 | 0.9982 | **0.9323** | 0.9917 | 0.9972 | 0.9972 | **0.9944** |
| w/o Attention | 0.8929 | **0.9696** | **0.9994** | 0.9297 | **0.9975** | **0.9983** | **0.9981** | 0.979 |
| w/o GeLU | 0.7058 | 0.7818 | 0.9931 | 0.7419 | 0.9864 | 0.9980 | 0.9961 | 0.9922 |

| | Dishwasher | | | | Washing Machine | | | |
|---|---|---|---|---|---|---|---|---|
| | **P** | **R** | **A** | **F1** | **P** | **R** | **A** | **F1** |
| Attention-bitcn | **0.8262** | **0.9921** | **0.9913** | **0.9016** | 0.5525 | **0.9950** | 0.9901 | 0.7105 |
| w/o Attention | 0.7978 | 0.9434 | 0.6495 | 0.1772 | 0.7154 | 0.9900 | 0.9951 | 0.8306 |
| w/o GeLU | 0.5412 | 0.9356 | 0.9657 | 0.6857 | **0.8136** | 0.9938 | **0.9971** | **0.8947** |

## 5. Conclusions

This work presents a temporal convolution model with the activation function GeLU and a residual structure for time-series load disaggregation based on bidirectional dilated convolution and multihead attention. The proposed model is compared with three related deep network models (i.e., CNN (S2P) [17], FCN [19], BitcnNILM [27]), LSTM, and BiL-STM on the public REDD and UK-DALE datasets. The experimental results show that

the proposed model achieves equivalent and even superior results for time-series load disaggregation and on/off state identification for individual appliances in NILM problems.

Whereas the operation of multihead self-attention increases the training time due to more weight parameters appended in the model, the fixed thresholds for the on/off state of the appliances cannot be adapted to the same types of appliances with different power consumption. In future work, we intend to decrease the model complexity when implementing edge computing, while improving its performance, e.g., by setting dynamic or optimal thresholds. In addition, the current supervised approach will be further developed into a generalized semisupervised or unsupervised approach.

**Author Contributions:** Conceptualization, Y.S. and X.Y.; methodology, Y.S., L.Z. and X.Y.; data curation, J.K.; validation, M.Z. and J.K.; investigation, Q.Y.; software, L.Z. and J.K.; writing—original draft preparation, Y.S.; writing—review and editing, X.Y., Y.S. and J.K.; visualization, L.Z. All authors have read and agreed to the published version of the manuscript.

**Funding:** This work was supported by the Technology Research Project of the National Grid of China (5700-202155204A-0-0-00).

**Data Availability Statement:** Not applicable.

**Conflicts of Interest:** The authors declare no conflict of interest.

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
