# Peer review of "Load Disaggregation Based on a Bidirectional Dilated Residual Network with Multihead Attention"

_electronics, doi:10.3390/electronics12122736_

Round 1
Reviewer 1 Report
In this paper the authors are dealing with the problem of load disaggregation. Given the total consumed power, they introduce a model with the aim of predicting the power consumption of the individual appliances.
The problem is both interesting and challenging. The paper has several strong points, however, other parts have to be strengthened.
Specifically:
1. The contribution of this work is unclear. The proposed model combines a bidirectional temporal convolution network with multi-head self-attention. However, several recent works have proposed very similar architectures:
* FJ Rendón-Segador, JA Álvarez-García, F Enríquez, O. Deniz, “Violencenet: Dense multi-head self-attention with bidirectional convolutional lstm for detecting violence”, Electronics, 2021.
* RA Hamad, M Kimura, L Yang, WL Woo, B. Wei, “Dilated causal convolution with multi-head self attention for sensor human activity recognition”, Neural Computing and Applications, 2021.
How is the proposed method different than the ones I indicate above? Is the usage of GeLU the only difference? It would be beneficial for the readership, if the authors included a list of contributions in the introduction.
2. Moreover, in many cases the use of English is bad. The text has multiple syntactical and grammatical errors which renders the paper hard-to-read, or even incomprehensible in some points. The first sentence of the Abstract is such an example. Problems of technical soundness are also present:
* The CNNs are Convolutional Neural Networks (not Convolution). Similarly, the layers are called convolutional, not convolution.
* We use the abbreviation ReLU for Rectified Linear Unit, not ReLu (line 8).
* In the Abstract, the authors describe the entire model in a single sentence (lines 6-10). A better structure is required in this case.
In general, the article must be revised to improve the quality of writing and eliminate all the syntactical and grammatical errors.
3. The experimental results indicate promising, but mixed results. The proposed attention-bitcn model outperforms all the others in the case of dish washers, but the other methods perform better on the other three appliances. So more experiments with additional datasets are required to provide concrete evidence about the usefulness of the attention-bitcn.
Additionally, I suggest that additional models should be included in the study. Especially LSTM and Bi-LSTM which have been proved effective in applications involving time-sequential data.
The quality of writing has to be improved as indicated in the original review.
Author Response
Response to Reviewer 1 Comments : (Please see the attachment)

Reviewer 2 Report
The paper deals with the topic of NILM which is interesting both from an academic as well as from an indutrial perspective. I have a few comments:
1. Where is such a solution expected to run? On the cloud? Maybe on the edge?
2. Also linked with 1. what about the computational effort/burden of the proposed implementation? How light is this solution but also how scalable it it is?
3. Regarding the references used in the Introduction section and the Literature Survey part, I would recommend to add more (and recent) papers in this field. An example: Athanasiadis, Christos L., Theofilos A. Papadopoulos, and Dimitrios I. Doukas. "Real-time non-intrusive load monitoring: A light-weight and scalable approach." Energy and Buildings 253 (2021): 111523.
4. Why do the authors focus only on specific appliances (WM, DW etc) and they do not extend their analysis to more energy intensive appliances, e.g. electric vehicles or HVAC?
5. How important is the data resolution for the proposed implementation? Could the authors apply this methodogy on data of different granularities?
Reviewer 3 Report
Dear authors below my comments:
1) Abstract section: You need to present some results from your experiments.
2) Lime 66: Elaborate more about functional bidirect TCN. Which threshold has been selected to consider an appliance ON/OFF;
3) Does the threshold has fixed value or dynamic?
4) line 72: Explain more about the meaning of dead neurons
5) Fig 2 is copy paste from reference [22]. You need to adopted or if you keep the same as is then you need to mention source:[22]
6) In Fig. 3, the dropout rate is missing.
7) Explain why you chose the same dataset and appliances from [22]
8) In your analysis, you must also consider the Seq2Seq model [16]
9) The term BitcNILM is not used in the literature review or in [22] reference. Change or elaborate more
10) In fig 1 change the term dilate rate to dilate factor/.
11) Why use multihead attention? Explain more
12) Explain why you selected only appliance type ON/OFF and not other appliances such as FSM or CVD.
13) In your analysis for appliance you can also include kettle and oven.
Dear Authors,
The English quality is good.
Round 2
Reviewer 1 Report
The authors have responded to my initial comments in a systematic and professional manner. I am now confident that the article can be accepted for publication on its present form.
Reviewer 2 Report
no further comments